# Whole genome landscapes of uveal melanoma show an ultraviolet radiation signature in iris tumours

Peter A. Johansson[1,20], Kelly Brooks[1,20], Felicity Newell[1], Jane M. Palmer [1], James S. Wilmott[2,3], Antonia L. Pritchard[1,4], Natasa Broit[1,5], Scott Wood[1], Matteo S. Carlino[2], Conrad Leonard[1], Lambros T. Koufariotis[1], Vaishnavi Nathan [1,5], Aaron B. Beasley [6], Madeleine Howlie[1], Rebecca Dawson[1], Helen Rizos [2,7], Chris W. Schmidt [1,8], Georgina V. Long[2,9], Hayley Hamilton[1,10], Jens F. Kiilgaard [11], Timothy Isaacs[12,13,14], Elin S. Gray [6,13], Olivia J. Rolfe[10], John J. Park[7], Andrew Stark[10], Graham J. Mann[2,15,16], Richard A. Scolyer [2,3,17], John V. Pearson[1], Nicolas van Baren[18], Nicola Waddell [1], Karin W. Wadt [19], Lindsay A. McGrath [10], Sunil K. Warrier[10], William Glasson[10] & Nicholas K. Hayward [1✉]

Uveal melanoma (UM) is the most common intraocular tumour in adults and despite surgical or radiation treatment of primary tumours, ~50% of patients progress to metastatic disease. Therapeutic options for metastatic UM are limited, with clinical trials having little impact. Here we perform whole-genome sequencing (WGS) of 103 UM from all sites of the uveal tract (choroid, ciliary body, iris). While most UM have low tumour mutation burden (TMB), two subsets with high TMB are seen; one driven by germline *MBD4* mutation, and another by ultraviolet radiation (UVR) exposure, which is restricted to iris UM. All but one tumour have a known UM driver gene mutation (*GNAQ, GNA11, BAP1, PLCB4, CYSLTR2, SF3B1, EIF1AX*). We identify three other significantly mutated genes (*TP53, RPL5* and *CENPE*).

[1] QIMR Berghofer Medical Research Institute, Brisbane, QLD, Australia. [2] Melanoma Institute Australia, The University of Sydney, Sydney, NSW, Australia. [3] Sydney Medical School, The University of Sydney, Sydney, NSW, Australia. [4] University of the Highlands and Island, Inverness, UK. [5] University of Queensland, Brisbane, QLD, Australia. [6] School of Medical and Health Sciences, Edith Cowan University, Joondalup, WA, Australia. [7] Department of Biomedical Science, Faculty of Medicine and Health Sciences, Macquarie University, Sydney, NSW, Australia. [8] Mater Research, Woolloongabba, QLD, Australia. [9] Department of Medical Oncology, Royal North Shore Hospital, St Leonards, Sydney, NSW, Australia. [10] Queensland Ocular Oncology Service, The Terrace Eye Centre, Brisbane, QLD, Australia. [11] Department of Ophthalmology, Rigshospitalet-Glostrup Hospital, University of Copenhagen, Copenhagen, Denmark. [12] Perth Retina, Perth, WA, Australia. [13] Centre for Ophthalmology and Visual Science, University of Western Australia, Crawley, WA, Australia. [14] Department of Ophthalmology, Royal Perth Hospital, Perth, WA, Australia. [15] Centre for Cancer Research, Westmead Institute for Medical Research, The University of Sydney, Westmead, Sydney, NSW, Australia. [16] John Curtin School of Medical Research, Australian National University, Canberra, Australia. [17] Royal Prince Alfred Hospital and New South Wales Health Pathology, Sydney, Australia. [18] Ludwig Institute for Cancer Research, Brussels, Belgium. [19] Department of Clinical Genetics, Rigshospitalet, Copenhagen, Denmark. [20]These authors contributed equally: Peter A. Johansson, Kelly Brooks. ✉email: nick.hayward@qimrberghofer.edu.au

Uveal melanoma (UM) arises from melanocytes in the uveal tract, and though less common than cutaneous melanoma, a higher proportion of UM patients die from the disease[1–3]. Risk determination of metastatic spread can be obtained through assessment of specific chromosome copy number alterations (CNAs)[4], gene expression profiles[5] and mutation status of known UM driver genes[6].

Previous genomic studies have pointed to the existence of four UM categories, strongly linked to prognosis[7–9]. Similarly we segregate our tumours into four categories based on CNAs: category 1 are chromosome 3 disomy (D3) tumours lacking chromosome 8q copy-number gain and frequently possessing *EIF1AX* mutations; category 2 are D3 UM with chromosome 6p and 8q gain and a high proportion of *SF3B1* mutations; category 3 are chromosome 3 monosomy (M3) tumours lacking chromosome 8q gain dominated by *BAP1* mutations; category 4 UMs are M3 with chromosome 8q gain and *BAP1* mutations. These genomic stratifications, while prognostic, are not indicative of treatment responses once progression has occurred.

To improve knowledge of UM genomics and to identify potential therapeutic targets, we conduct a whole-genome sequencing (WGS) study of 103 UM, comprised of 91 primary tumours and 12 metastases, with matched germline DNA. Eighty-four tumours originate from the choroid, eight from the iris, four from the ciliary body, and seven without known primary uveal site (Supplementary Data 1).

## Results

**Recurrent copy number aberrations.** In line with previous studies[8–11], TMB was low in the majority of UM (median 0.50 mutations per megabase, range 248–42,669, Supplementary Data 2) and tumours generally displayed low counts of structural variations (SVs) (median: 13; range 0–213) (Fig. 1). One sample had noticeably more SVs, of which the majority (71%) were midsized (<100 kb) deletions, suggesting this was not due to chromothripsis. There were no additional notable features or known driver mutations in this sample. Commonly observed losses of chromosome 1p, 3, 8p, 6q and 16q were present, as were gains of 1q, 6p and 8q (Fig. 2)[4,12]. Two samples presented with whole-genome duplication (WGD). Tumours were grouped into the four categories described. Category 4 was predominant, with 55 samples (53%) displaying M3 and copy-number gain of chromosome 8q. In previous studies[8,9], the vast majority (93–95%) of M3 tumours also displayed 8q gain, whereas in this cohort a notable proportion (13/68, 19%) of the M3 samples showed no 8q gain (category 3) (two-sided Fisher's exact test, $P = 0.04$). Some D3 samples (11/35, 31%) had gain of 8q (category 2), as previously observed[8,9]. Chromosome 8p loss was only observed in samples with 8q gain (categories 2 and 4), reflecting the formation of isochromosome 8q, whereas the other alterations were spread across categories 2, 3 and 4. The majority of category 1 tumours lacked these common, large CNAs, instead displaying either few or more dispersed rearrangements.

**UV mutation signatures in iris tumours.** Assessment of single base substitution (SBS) signatures revealed SBS5 to predominate in most cases, with strong representation of SBS3, SBS39 and SBS40 in some samples (Fig. 1)[13]. Two samples were dominated by mutation signature SBS1 (associated with spontaneous deamination) and had correspondingly high TMB (>3 mutations per Mb). As previously observed[14,15], these features corresponded to the presence of germline loss-of-function (LOF) *MBD4* mutations; this takes the published tally of germline *MBD4* mutant UM cases to six[14,15], strengthening its role as a UM predisposition gene. The two UMs with germline LOF *BAP1* mutations

displayed no unique features. Strikingly, all iris melanomas displayed the genomic features associated with UVR damage (mutation signatures SBS7a, SBS7b and DBS1[13] combined with a high TMB). While exposure to UVR has been suggested as a cause of the elevated UM risk among arc-welders[16], no molecular evidence of UVR as an aetiological factor has yet been observed in UM sequencing studies. The iris is located anteriorly within the uveal tract and is directly exposed to sunlight that breaches the cornea; we now show that UVR-associated DNA damage results from this exposure and is a unique genomic feature of iris UM.

**Patterns of driver mutations and chromosomal aberrations.** Assessment of known UM driver genes revealed an oncogenic driver mutation in 102 of 103 tumours: 51 in *GNAQ* (48 p.Q209P/L, two p.R183Q, one p.G48L), 46 in *GNA11* (44 p.Q209L/P, two p.R183C), five in *PLCB4* (three p.D630Y, two p.D630N) and two in *CYSLTR2* (p.L129Q). These mutations were generally mutually exclusive except for two PLCB4 p.D630 mutations that co-occurred with GNAQ/GNA11 p.R183H mutations. This co-occurrence between PLCB4 mutation and the minor hotspot p.R183, rather than the stronger oncogenic p.Q209 hotspot mutations, has previously been described in the UM TCGA data[8]. Though not previously highlighted, GNAQ p.G48L mutations have been reported in two UM samples from two separate studies[8,17], as well as in two hepatic small vessel neoplasms, which are driven by activating *GNAQ/GNA14* mutations[18]. This suggests that GNAQ p.G48L is another minor UM oncogenic hotspot mutation. Similar to previous observations[19,20], *BAP1* was the most altered gene in M3 samples (75%), including eight splice site mutations, two germline and 32 somatic LOF mutations, and three cases with disrupted *BAP1* due to SV breakpoints. In addition, two D3 tumours carried *BAP1* mutations, indicating that although *BAP1* inactivation typically occurs after M3[8], *BAP1* aberration can also occur in D3 tumours, which may or may not later undergo loss of chromosome 3. Of note, one of these D3 tumours (MELA_0800) had a low *BAP1* variant allele frequency (VAF = 9/80) suggesting it was only present in a subclone, and as copy number tools are not as sensitive as mutation callers, it is possible that the subclone had loss of heterozygosity (LOH) that was not detected by the algorithm. Five tumours had *BAP1* mutations and copy-neutral LOH, suggesting that the mutations occurred before WGD in the two tetraploid UMs and before the LOH event in the three diploid UMs. *SF3B1* mutations were present in 15 tumours, the majority occurring in category 2, in line with other studies[7–9]. *EIF1AX* hotspot mutations were observed in 19% of tumours. *EIF1AX* mutations were first discovered in D3 UMs[21] and in the TCGA cohort they were restricted to category 1 tumours (D3 and no 8q gain)[8], while in the cohort presented by Royer-Bertrand and colleagues two of seven mutations were seen in tumours with M3 and/or 8q gain[9]. Similarly, here six of the 20 *EIF1AX* mutations (30%) were seen in UM with M3 ($n = 5$) or 8q gain ($n = 1$) (Fig. 3a).

**Significantly mutated genes.** In addition to these known UM genes, three other statistically significantly mutated genes (SMGs) were identified (*CENPE, TP53, RPL5*; Supplementary Table 1). Three missense mutations and two LOF SVs were identified in *CENPE*, along with one LOF germline mutation (late truncating) (collectively ~5% of UM). An additional six samples were hemizygous for *CENPE* (Fig. 3a). CENPE is a plus end-directed kinetochore motor protein which plays a critical role in mitosis and chromosome segregation. Knockdown of CENPE has been shown to cause chromosome misalignment and lagging[22,23]. Two of the missense mutations (p.R14W and p.R251W) occurred in the kinesin motor domain (Fig. 3c) at highly evolutionarily

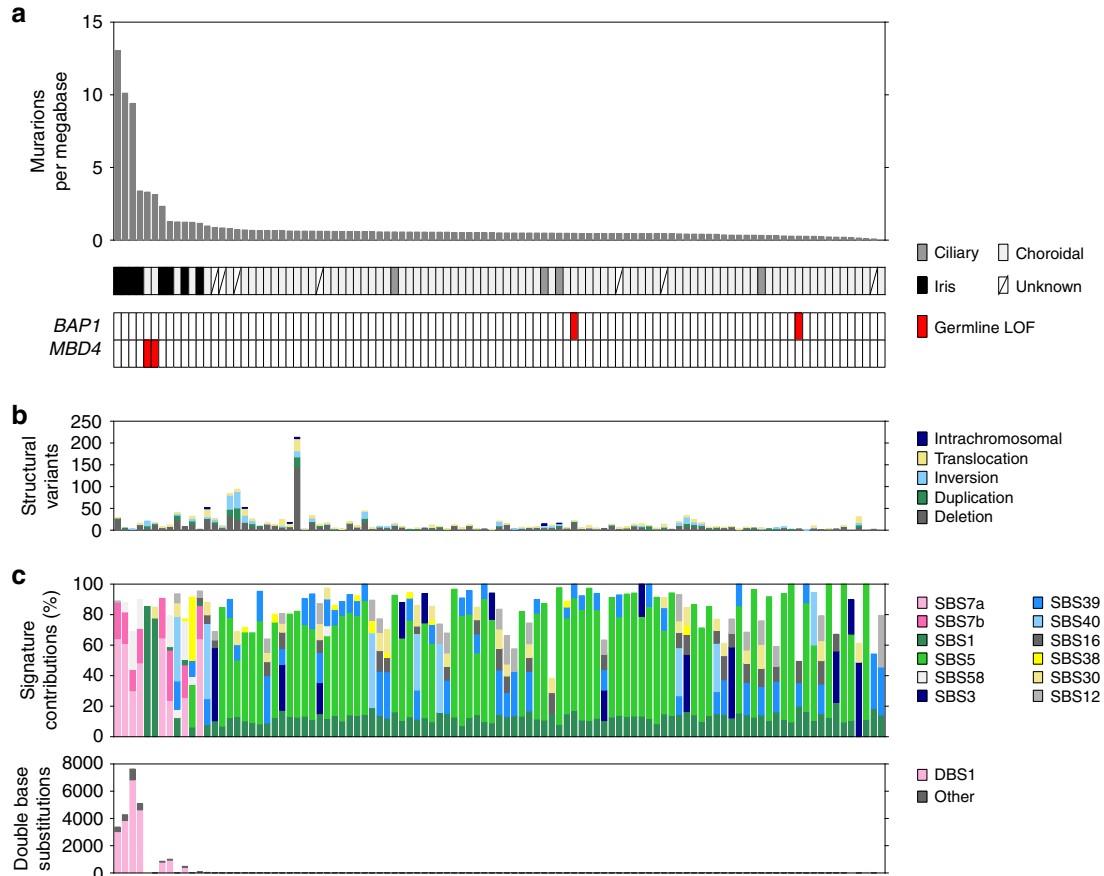

**Fig. 1 Mutational landscape of 103 UM tumours. a** Tumour mutation burden (substitutions and indels) in 103 uveal melanomas. **b** Burden of structural variation in each tumour displayed as numbr of events. Variants are coloured according to their event category. One outlier was observed with >200 events, mostly mid-sized (<100 kb) deletions. **c** Spectrum of single base substitution (SBS) signatures shown as percentage. Most samples were dominated by signature 5, which has been observed in most cancers. Signatures SBS7a and SBS7b, only seen in iris melanomas here, are commonly observed in cutaneous melanoma and associated with exposure to ultraviolet radiation (UVR). SBS1 was predominantly observed in melanomas with loss of MBD4 and is associated with deamination of 5-methylcytosine. Spectrum of double base substitutions shown as number of substitutions in each tumour. DBS1 is characterised by CC>TT transition and associated with exposure to UVR.

conserved regions (ECRs) and the third (p.Q1098P) occurred in a reasonably conserved residue (Supplementary Fig. 1). An additional two missense mutations were identified in the UM TCGA cohort at p.I1038T (weak ECR) and p.K1821N (reasonable ECR), both also in a coiled-coil domain[24]. CENPE has been shown to interact with CENPF, BUB1B and Aurora B, the latter two being critical in the activation of the spindle assembly checkpoint[25–27]. In the UM cohort described here, one sample had a BUB1B missense substitution (p.R691H) within the region reported to directly interact with CENPE[25]; another had a p.D303E substitution in a highly ECR of the Aurora B catalytic domain. It is possible that disruption of this pathway is responsible for creating genomic instability allowing for chromosome aberrations to occur. Indeed, the twelve UM with *CENPE* alterations had significantly higher genome percentages with CNAs (Mann–Whitney, $P = 0.028$, median 23% vs 15%), though this association is confounded since tumours with high CNA generally have more genome-wide regions of LOH. Studies both in cell line and mouse models have demonstrated that CENPE/Cenpe functions in a haploinsufficient manner, with elevated levels of chromosome missegregation observed in heterozygous cells and animals[28,29]. Functional work on CENPE missense mutations is required to determine their impact in UM.

**TP53 is commonly disrupted in uveal melanoma**. High expression of p53 has been reported in some UM, often associated with histological and clinicopathological features correlated with poor prognosis, but the potential genetic basis of these observations was not assessed[30–32]. Somatic mutations in *TP53* have been described in two UM. A hotspot mutation p.R175H ($n = 1216$ in IARC *TP53* database) was observed in a hypermutated metastatic UM with deficient MBD4[33] and another hotspot mutation p.M237I ($n = 196$ in IARC *TP53* database) was observed in a UM in a pan-cancer study of metastatic tumours[17]. Here we identified *TP53* as an SMG and report six somatic *TP53* mutations across four tumours in addition to eight cases of LOH (Figs. 3a, b). One LOH case overlapped with an LOF mutation (p.C277*) resulting in a double-hit in *TP53*. Another double-hit was seen in a sample with two missense mutations (p.H193R and p.T155I) confirmed as occurring on different alleles by assessing read pairs spanning both mutations. To evaluate the consequence of these mutations, we applied a computational prediction tool, FATHMM[34], and assessed the results of two comprehensive characterisation studies of *TP53* mutations (Table 1)[35,36]. p.H193R is a recurrent hotspot classified as pathogenic by PHANTM, RFS and FATHMM, while the consequence of p.T155I is more uncertain, as the variant is classified pathogenic by PHANTM but predicted to have neutral impact by RFS and

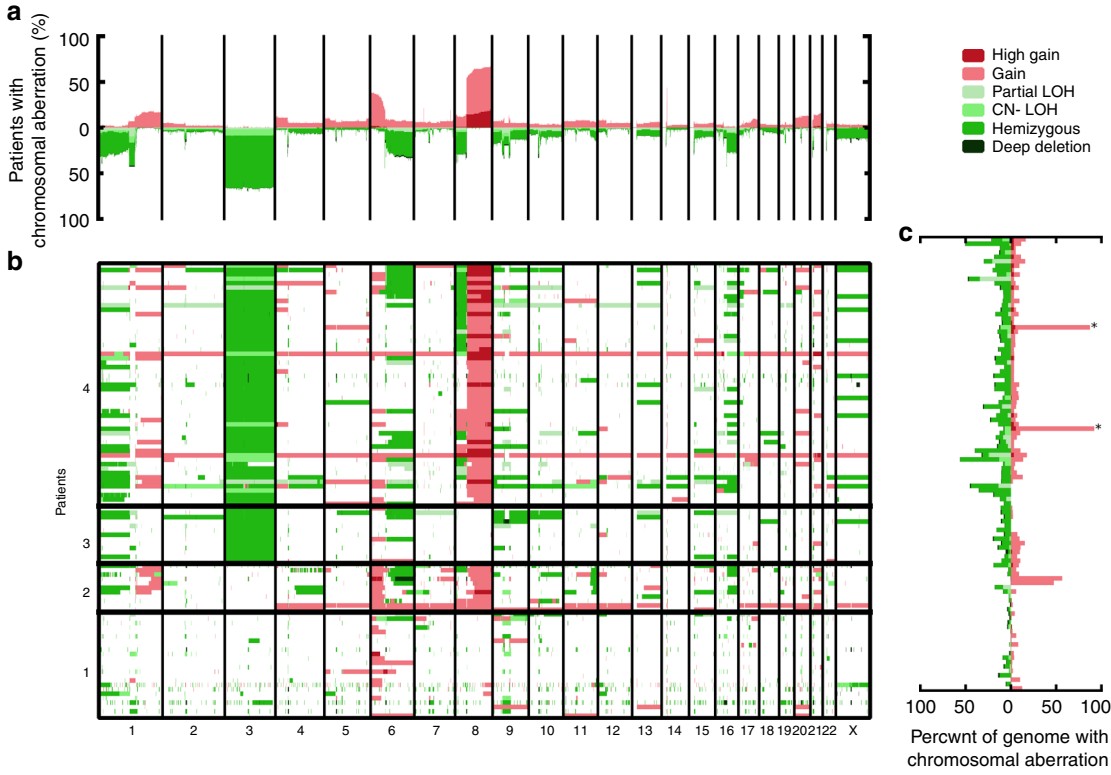

**Fig. 2 Copy number gains and losses in uveal melanoma.** Samples are arranged according to their category defined by status of chromosome arms 3p and 8q. **a** Total number gains/losses in each genomic region. **b** Gains (red) and losses (green) shown in each sample and region. **c** The fraction of the genome having aberrations in each sample. * Sample with whole-genome duplication. Source data are provided as a Source data file.

FATHMM. Finally, one UM had a LOF mutation (p.R342*, COSM11073) and a missense p.R248Q mutation, both of which frequently occur in malignancies and are classified as pathogenic (Table 1). RNA-seq data revealed one read pair spanning both positions, which contained p.R248Q and was wildtype for p.R342; furthermore, there was a significantly lower VAF at p.R342 (4/78) than at p.R248 (25/68) (two-sided Fisher's exact test, $P = 2 \times 10^{-6}$). These data suggest the two mutations occurred on different alleles, with the majority of the transcripts from the p.R342* allele undergoing nonsense mediated decay. *TP53* is a tumour suppressor gene frequently deleted or mutated, resulting in either no production of p53 or the expression of a truncated and unstable protein. The spectrum of a few highly recurrent missense mutations, including p.R248Q, has, however, given rise to hypotheses that these hotspot mutations translate to mutant p53 with gained oncogenic functions[37]. For example, the p.R248Q mutation reported here has been shown to increase the migratory potential of cells in an in vitro model[38].

**RPL5 is significantly mutated in uveal melanoma.** SMG analysis also identified *RPL5*, with truncating mutations in three cases and LOH in an additional 30 cases (Fig. 3a). No UM had a double hit in *RPL5*, in line with previous studies suggesting *RPL5* is a haploinsufficient tumour suppressor, with heterozygous inactivation in glioblastoma (11%), breast cancer (34%) and cutaneous melanoma (28%)[39]. In UM the majority of *RPL5* LOH occurred through loss of chromosome 1p, where the gene is located, but in four cases, focal loss occurred, suggesting that *RPL5* may drive positive selection for 1p loss. Chromosome 1p loss in UM has been reported as a marker of poor prognosis, independent of M3[12]. *RPL5* encodes ribosomal protein L5, which complexes with 5S rRNA and forms an important part of the impaired ribosome biogenesis checkpoint (IRBC). Together with RPL11 and ARF,

RPL5 binds to and inhibits MDM2, resulting in p53 stabilisation in response to blocks in ribosome biogenesis and nucleolar stress[40–42]. Of note, one truncating mutation in *RPL11* was observed in the UM TCGA cohort[8] further supporting the importance of this pathway in UM. Given the link between RPL5 and p53, we tested for an association between aberrations in *RPL5* and *TP53*. Indeed, mutations in these genes were mutually exclusive, but when also considering copy-number loss there was no association. However, as p53 functions in multiple pathways, it may be inactivated in some tumours with defective RPL5 due to selective pressures outside the IRBC response. Interestingly, the IRBC is often triggered by oncogene-induced translational stress[40–42], with oncogenic *MYC* being shown to significantly increase IRBC activation[41–44]. In addition to their IRBC role, RPL5 and RPL11 have also been shown to bind to *MYC* transcripts, mediating RNA-induced silencing and interfering with c-Myc driven transcription[45,46]. Overexpression of *MYC* is thought to be the driving force behind positive selection of chromosome 8q gains in UM. It is possible that c-Myc overexpression leads to ribosomal stress and IRBC activation, likely inhibiting tumour growth. To overcome this, it may be necessary to disrupt this pathway through mutation/loss of either *RPL5*, *RPL11* or *TP53*. Supporting this notion, RPL5 and TP53 disruption (including mutations, SV breakpoints and chromosomal copy loss) was more common in cases with 8q gain (48%) than in cases without 8q gain (22%) (two-sided Fisher's exact test, $P = 0.01$). Tumours with alterations in this pathway were predominantly M3, making it difficult to disentangle the prognostic impact; however, these individuals had poorer prognosis (Supplementary Fig. 2).

**Genomic categories correlate with prognosis.** To correlate UM genomic categories with prognosis, time from first presentation to metastasis was examined for all patients. Comparing the four

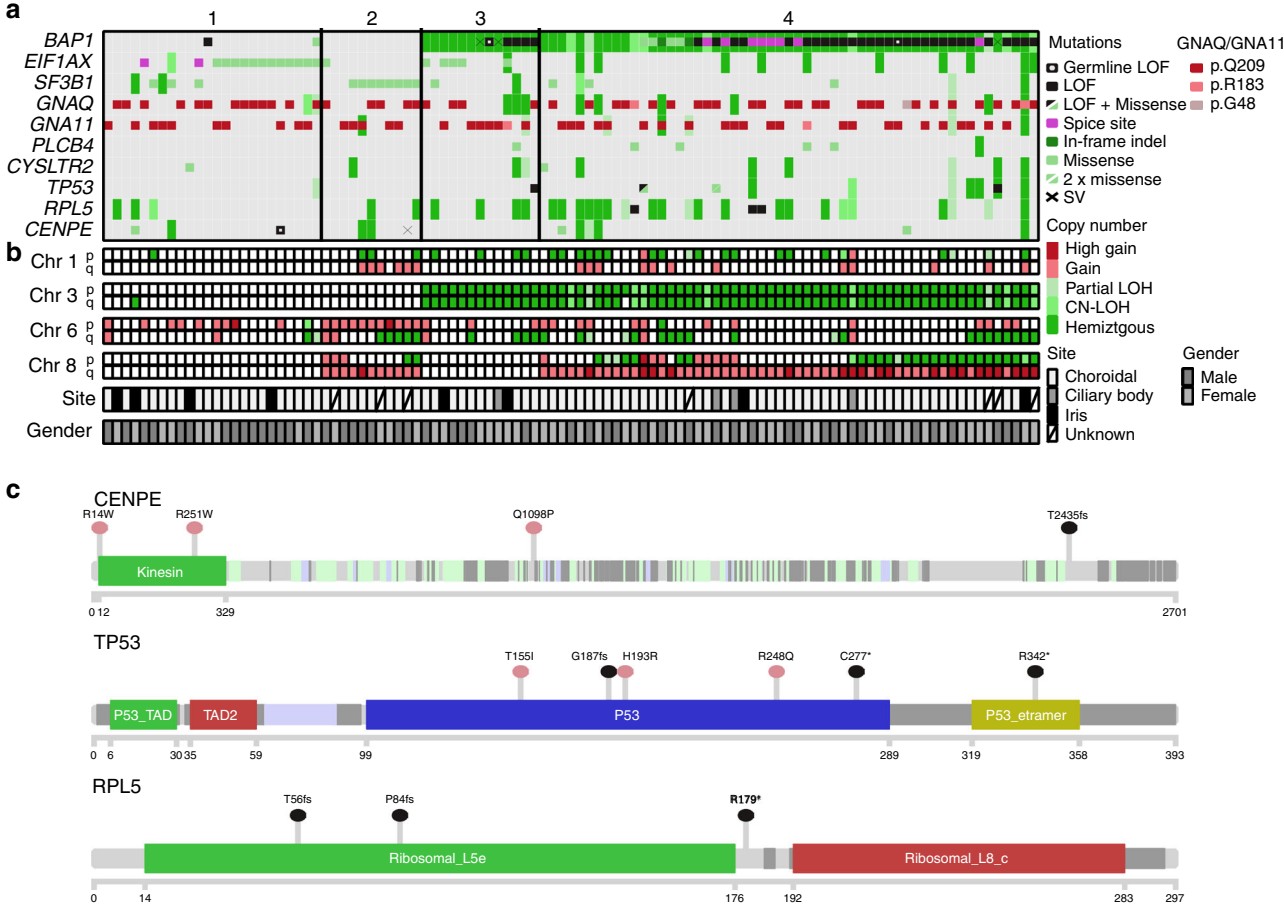

**Fig. 3 Significantly mutated UM genes.** Samples are arranged in the same order as Fig. 2 by their category (1–4). **a** Alterations (substitutions, indels, SVs and CNAs) in significantly mutated genes and known UM driver genes. Nonsense mutations and frame-shift indels are labelled LOF. Loss-of-function breakpoints due to structural variants are labelled SV. **b** Copy number status of the commonly aberrant chromosomes 1, 3, 6 and 8, in UM. **c** Positions of mutations in encoded protein of significantly mutated genes, *CENPE*, *TP53* and *RPL5*. Mutations are coloured black (LOF) and green (missense). Proteins domains are depicted wider and structurally important regions are coloured dark grey (disorder), light green (coiled-coil) and light blue (low complexity).

| Table 1 Classification of *TP53* mutations. | | | | | |
|---|---|---|---|---|---|
| Variant | Effect | PHANTM[a] | RFS[b] | FATHMM[c] | Count[d] |
| p.R342* | Nonsense | 0.70 | na | 0.73 | 96 |
| p.R248Q | Missense | 0.81 | −0.04 | 0.98 | 946 |
| p.C277* | Nonsense | 0.42 | 0.01 | 0.96 | 9 |
| p.H193R | Missense | 1.48 | 0.38 | 0.99 | 101 |
| p.T155I | Missense | 0.97 | −1.13 | 0.39 | 21 |
| p.G187fs | Frameshift deletion | na | 0.48[e] | na | 5 |

*PHANTM* phenotypic annotation of TP53 Mutations, *RFS* relative fitness score, *FATHMM* functional annotation through Hidden Markov Models.
[a]The score is 0 for common benign polymorphisms and 1 for recurrent somatic hotspot mutations.
[b]The score is on average −2.50 for synonymous variants and 0.42 for protein truncating variants.
[c]Scores above 0.5 are considered deleterious.
[d]Number of somatic cases in the IARC TP53 database.
[e]The exact nucleotide variant was not characterised in RFS; the average score of LOF variants at codon 187 is shown.

TCGA categories, there was a trend that category 4 tumours had shorter relapse-free survival (RFS) (median: 2.5 years) than UMs in category 3 (median: 7.5 years) and similarly category 2 had shorter RFS (median: 7.0 years) than category 1 (median not reached), but the differences were not statistically significant

($P_{3vs4} = 0.11$, $P_{1vs2} = 0.28$) (Fig. 4a). However, M3 UM had significantly shorter RFS (median: 2.9 years) than D3 UM (median 7.0 years, log-rank test, $P = 0.001$, Fig. 4b), confirming the prognostic strength of M3/D3 status. Interestingly, while iris melanomas are associated with earlier detection and favourable prognosis, 4/8 iris cases had M3, with two already having progressed to metastatic disease. The high TMB in iris UM reported here suggests they are more likely to respond to immunotherapy, given the observations for *MBD4* germline UM patients[14,15], who have high TMB, a predictor of response to immunotherapy response in cutaneous melanoma and other cancers[47]. In combination with previous reports of metastatic iris UM[48], these data suggest a subset of iris UM are at high risk for disease progression and will likely respond to immunotherapy in the event of progression and perhaps even in the adjuvant setting.

## Methods

**Human melanoma samples.** Fresh-frozen tissue and matched normal samples were obtained from the Terrace Eye Centre (Brisbane, Australia), Rigshospitalet (Copenhagen, Denmark), Melanoma Institute Australia (Sydney, Australia), Lions Eye Institute (Perth, Australia), Royal Perth Hospital (Perth, Australia), St John of God Hospital (Subiaco, Australia), and Ludwig Institute for Cancer Research (Brussels, Belgium). All tumour and blood/saliva samples were accrued with written informed patient consent following institutional review board approval from the Human Research Ethics Committees of the QIMR Berghofer Medical Research Institute, the Sydney Local Health District RPAH zone, the University of

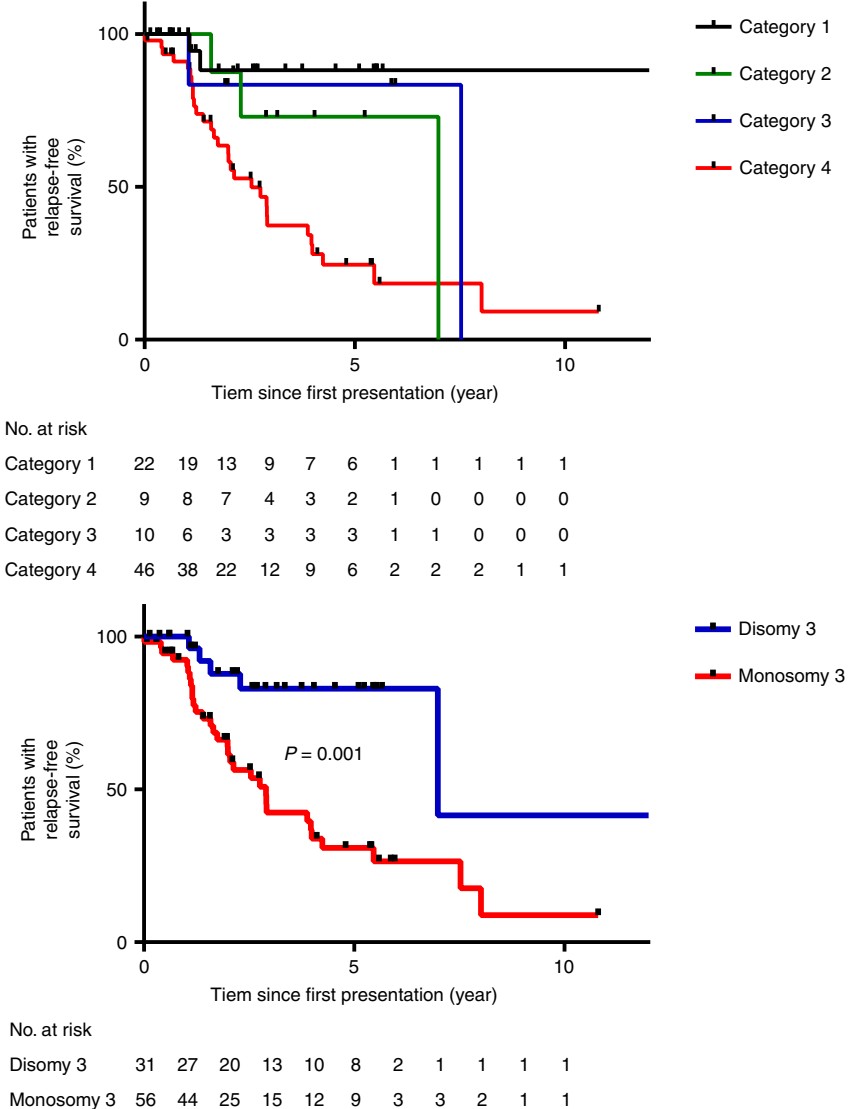

**Fig. 4 Kaplan–Meier estimates of relapse-free survival (RFS) for UM patients.** Higher risk categories have shorter RFS. UM patients with monosomy of chromosome 3 has significantly shorter RFS than patients with disomy 3 (blue). Difference between curves were assessed using a two-sided log-rank (Mantel-Cox) test; $p$-values were not adjusted for multiple testing.

Western Australia, the Capitol Region of Denmark and the Ludwig Institute for Cancer Research. Patient information was stored in Progeny 10.

**DNA extractions**. Tumour DNA was extracted using the AllPrep DNA/RNA Kit (80204, Qiagen Ltd, Hilden, Germany), blood DNA using standard salting out methods, and saliva DNA was collected and extracted using the Oragene DNA kit (OG-500, DNA Genotek, Ottawa, Canada) according to the manufacturer's instructions. All samples were quantified using a NanoDrop (ND1000; Thermo Fisher Scientific, Waltham, Massachusetts, USA) and Qubit dsDNA HS Assay Q32851; Life Technologies, Carlsbad, California, USA).

**Whole-genome sequencing**. Sequencing libraries were constructed using TruSeq DNA Sample Preparation kits (Illumina, San Diego, California, USA) according to the manufacturer's instructions. WGS was performed on Novaseq or HiSeq X Ten instruments (Illumina) by Macrogen (Seoul, South Korea). Sequence data were adapter trimmed using Cutadapt v1.9[49] and aligned to the GRCh37 assembly using BWA-MEM v0.7.12 and SAMtools v1.8[50,51]. Duplicate reads were marked with Picard MarkDuplicates v1.129 (https://broadinstitute.github.io/picard).

Similarly, RNA sequence reads were adapter adapter trimmed using Cutadapt and aligned using STAR v2.5.2a to the GRCh37 assembly with the gene, transcript, and exon features of Ensembl gene model v70[52].

**Somatic mutations**. Somatic SNV and indels were detected using an established pipeline[53] in which SNVs were called with qSNP[54] and GATK HaplotypeCaller

and indels were detected with GATK[55]. The contribution from different mutation signatures was inferred by approximating (minimising the squared error) the distribution of mutations as a linear combination of COSMIC signature v3[13] with the constraint that contributions were non-negative. SMGs were identified using MutSigCV 1.3.5 (via GenePattern) as well as Oncodrive-fm and OncodriveClust (via IntOGen)[56]. A Benjamini-Hochberg adjusted $p$-value ($q$-value) below 0.05 was considered significant. To avoid false negatives in hotspots of known UM genes, these regions (GNAQ p.48, p.183, and p.209; GNA11 p.183 and p.209; SF3B1 codons p.625, p.666 and p.700; EIF1AX codons p.1-20, PLCB4 p630; and CYSLTR2 p.129) were called with higher sensitivity. For each sample and genomic position the variant and reference read counts were compared with the variant and reference read counts in the pool of all 103 normal/germline samples at that specific position. Fisher's exact test was used to identify somatic mutations and a Bonferroni corrected $p$-value below 0.001 was considered statically significant.

**Classification of TP53 mutations**. To evaluate the *TP53* mutations, they were compared with two comprehensive characterisation studies. The Phenotypic Annotation of TP53 Mutations (PHANTM) score v1.0 is a weighted sum of z-scores for which common (i.e. benign) germline variants have values around 0 and recurrent somatic hotspot mutations have scores around 1[35]. The relative fitness score (RFS) is on average −2.50 for synonymous variants, while the average score for protein truncating variants is 0.42[36]. FATHMM is an in silico tool predicting the probability that variants are deleterious[34]. Scores above 0.5 are considered deleterious. The IARC *TP53* mutation database (R20, July 2019) was used to evaluate how frequent mutations are[57].

**Copy number aberrations and SV**. Copy number aberrations were identified using ascatNgs[58]. Copy number of at least 6 was considered high gain. The underlying model of ascatNgs assumes the data come from two clones of cells: the tumour and normal contamination. For a heterogeneous tumour it may therefore overestimate copy numbers; to distinguish heterogeneity from WGD, the distribution of VAF for somatic mutations within regions with allelic balance were used. For a tetraploid tumour, two peaks in VAF are expected corresponding to mutations occurring before and after the copy number gain, with the latter having half the VAF of the former. Statistical models for the data coming from a homogeneous tetraploid tumour and from a heterogeneous diploid tumour, respectively, were inferred using maximum-likelihood and if the heterogeneity was significantly more likely ($P < 0.001$), copy numbers were adjusted.

Structural variants were identified using an in-house tool, qSV, as previously described[53]. Gene truncating breakpoints and consequence of the SVs were determined using in-house scripts and transcript annotation from Ensembl.

**Survival analysis**. Relapse-free curves were estimated using the Kaplan–Meier method. Difference between curves was assessed with the log-rank (Mantel-Cox) test.

**Reporting summary**. Further information on research design is available in the Nature Research Reporting Summary linked to this article.

## Data availability

The BAM files are deposited in the European Genome-phenome Archive ([https://www.ebi.ac.uk/ega/]) with accession number EGAS00001001552. The source data underlying Fig. 2 are provided as a data source file. All other data are available in the Article, Supplementary Information or available from the author upon reasonable request.

## Code availability

Tools used in this publication that were developed in-house are available from the SourceForge public code repository under the AdamaJava project ([http://sourceforge.net/projects/adamajava/]). Updated versions of software are available at [https://github.com/AdamaJava].

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

## Acknowledgements

We are indebted to the patients and their families for their participation and support of this study, and the many clinicians and allied health professionals involved in their management. We gratefully acknowledge the participation of Sharon Morris, Nicholas O'Rourke, David Cavallucci Thomas O'Rourke, Helen Marfan and Rachel Susman. As well, we thank Kevin Whitehead, Gary Quagliotto, and Sullivan Nicolaides Pathology staff for processing the Queensland samples. This project was funded by the National Health and Medical Research Council (NHMRC; 1093017), the Walking On Sunshine Foundation, Anne Stanton, Nicola Laws and Lloyd Owen in Memorial and Civic Solutions. This study was also funded by Fight for Sight, Denmark. A.L.P. is supported by Highland Island Enterprise (HMS9353763). This work was supported by an NHMRC Program Grant (G.V.L., G.J.M., R.A.S. and N.K.H.). G.V.L. is supported by an NHMRC Practitioner Fellowship and The University of Sydney, Medical Foundation. R.A.S. is supported by an NHMRC Practitioner Fellowship. Support from Melanoma Institute Australia and The Ainsworth Foundation is also gratefully acknowledged. J.S.W. is supported by a NHMRC early career fellowship (1111678). N.W. is supported by an NHMRC Senior Research Fellowship (1139071). N.K.H. is supported by an NHMRC Senior Principal Research Fellowship (1117663).

## Author contributions

P.A.J. and K.B. were responsible for data interpretation and writing the manuscript. N.K.H., R.A.S. and G.J.M. conceptualised the study. P.A.J., F.N., C.L., S.W., N.B., L.T.K., J.V.P. and N.W. performed data analysis. K.B. N.B., V.N., C.W.S., R.D. and A.B.B. were responsible for sample processing and extraction. R.A.S., N.K.H., R.A.S., G.V.L., H.R., E.G. acted in a supervisory capacity. A.L.P., J.M.P., J.S.W., M.S.C., M.H., H.R., G.V.L., H.H., J.J.P., O.J.R., J.F.K., T.I., N.vB., K.W.W., L.A.M., A.S., S.K.W., W.G. were responsible for patient data curation, recruitment and tumour acquisition. All authors reviewed and edited the manuscript.

## Competing interests

J.V.P. and N.W. are founders and shareholders of genomiQa Pty Ltd, and members of its Board. K.W.W. participated in one Advisory board meeting for MSD and AstraZeneca. R.A.S. receives fees for professional services from Merck Sharp & Dohme, GlaxoSmithKline Australia, Bristol-Myers Squibb, Dermpedia, Novartis Pharmaceuticals Australia Pty Ltd, Myriad, NeraCare and Amgen. G.V.L. is consulant advisor for Aduro, Amgen, Array, BMS, MERCK MSD, Novartis, Pierre-Fabre, Roche. None of these relationships involve the work described in this manuscript. The remaining authors declare no competing interests.
