## [Peer Review File · Nature Communications]

Reviewers' comments:

Reviewer #1 (Remarks to the Author):

In this manuscript the authors describe results of whole-genome sequencing (WGS) of 103 uveal melanomas (UM). They confirmed published data showing low mutation burden, frequent mutation in known UM genes (GNAQ, GNA11, BAP1, PLCB4, CYSLTR2, SF3B1, EIF1AX), and commonly observed losses and gains of chromosomes. The new information includes high mutation rates in the iris UM harboring UV signatures, and mutations in three genes (TP53, RPL5 and CENPE). In general, the authors confirmed previous genomic data on uveal melanoma with marginal new observations. It is interesting that the cohort included large number of primary tumors (n=91), and the expectation is to identify clonal evolution of metastatic competence, and yet none of that is described. As Harbour et al summarized in one of their publications "This implies that the metastatic proclivity of UM is "set in stone" early in tumor evolution" (Field, M. G., Durante, M. A., Anbunathan, H., Cai, L. Z., Decatur, C. L., Bowcock, A. M., Kurtenbach, S., and Harbour, J. W. (2018) Punctuated evolution of canonical genomic aberrations in uveal melanoma. *Nat Commun* 9, 116)

This manuscript belongs to a more specialized journal such as the *Am J Hum Genet* in which a similar article was published in 2016 (PMC5097942).

Minor comment:

Red in Figure 2 should be used for the more conventional copy number gain rather than loss to avoid reader confusion.

Reviewer #2 (Remarks to the Author):

In this manuscript, Johansson and Coll provide an analysis by WGS of the largest series of uveal melanoma (103 UM cases) described so far. This analysis largely confirms previously published results, and further shows the presence of UVR mutational signature in iris UM, and some evidence of new drivers in some cases. It is difficult to understand the interest of using WGS rather than WES, when the authors are not exploiting their WGS data to extract additional information such as structural variants or alternations of non coding regions, such as promoter regions.

Major comments

- 1) Title is not supported by evidence. If UVR signature is shown in iris UM, no causal link with the disease is provided. UVR signature also exists also in normal melanocytes or keratinocytes!
- 2) L70 "One sample had noticeably more SVs, of which the majority (71%) were mid-sized (<100 kb) deletions, suggesting this was not due to chromothripsis" Limited information is provided for this interesting outlier case. Any correlation of driver mutations?
- 3) L195 "In addition, two D3 tumours carried BAP1 mutations, indicating that although BAP1 inactivation typically occurs after M3, BAP1 aberration can also occur in D3 tumours, which may or may not later undergo loss of chromosome 3." These observations are not as surprising as they appear in the text. One is associated with a partial loss of chr3, which is relatively rare but previously reported, and for the second case, BAP1 mutation seems to have emerged as a subclonal event (low VAF), in which the status of chr3 is yet to be determined.
- 4) Evidence for new drivers in UM are scarce:
 - "three additional significantly mutated genes" but only one algorithm (Oncodrive-fm) shows this significance, and with a marginal p-value for CENPE.

- CENPE: the second wild-type allele is retained with all variants
L130 "It is possible that disruption of this pathway is responsible for creating genomic instability allowing for chromosome aberrations to occur." The authors should have all data to answer this question. Any correlation with WGD ? with number of CNA ? The authors should take into account that Oncodrive-fm does not correct for the size of the gene, and that CENPE is a very large gene (so it is highly likely that these variants are passenger).
- TP53: "Mutations in TP53 are common in many cancers but have not been described in UM to date."
False: see Hajkova et al Sci Rep 2018 and ref, Rodrigues et al CCR 2019. One of these cases is an hypermutated metastasis, which should be mentioned. Actually, the bottom line is that two cases are inactivated for TP53, one hypermutated metastasis (as in Rodrigues et al CCR 2019) and one primary of unknown origin.
- RPL5 : None of these mutations are associated with LOH. One case is hypermutated.
L151 "RPL5 along with RPL11 and ARF are responsible for inhibiting MDM2 and stabilising p53 in response to blocks in ribosome biogenesis and nucleolar stress, often triggered by oncogene-induced translational stress. Supporting the importance of this pathway in UM, TP53 and RPL5 were altered in 42% of cases in this cohort." This is an overstatement supported by no evidence, but that the longest chromosome (chr1) is frequently altered in UM.
- 5) L167 "Interestingly, while iris melanomas are associated with earlier detection and favourable prognosis, 4 of 8 iris cases had M3, with two already having progressed to metastatic disease". None of these metastatic patients received immunotherapy ?

Minor comment

- The first publication of of UM characterized by WGS (Furney Cancer Discov 2013) should be cited
- L97 "These mutations were generally mutually exclusive except for two PLCB4 p.D630 mutations. As seen in the UM TCGA cohort, overlap occurred with the minor GNAQ p.R183Q/ GNA11 p.R183C hotspots rather than the stronger oncogenic p.Q209 hotspot mutations." Confusing: something like "two PLCB4 p.D630 co-occurred with GNA p.R183" would be more straightforward.
- L145: "These data suggest the two mutations are biallelic". No, the authors suggest composite heterozygous mutations or mutations in trans.

Reviewer #3 (Remarks to the Author):

Johansson et al describe the whole genome landscapes of 103 uveal melanoma patients, including 91 primary tumors and 12 metastases. The overall conclusions of the study are: (1) TMB is low in the vast majority of uveal melanoma patients, consistent with prior studies; but two patients with loss of function mutations in MBD4 with higher TMB. Similarly all iris-derived melanomas had high UVR damage signatures, suggesting that this subset might have unique biological properties. The finding that iris UM have higher TMB has not been reported, though the MBD4 mutations have been previously described (as cited by the authors)

(2) A mutation in GNAQ pG48L appears to be a minor oncogenic hotspot mutation

(3) Multiple aberrations in TP53 pathway. However, functional data for most of these mutations could be strengthened by evaluating recent, comprehensive analysis of p53 mutations (e.g. Giacornelli et al, Kotler et al, etc).

(4) RPL5 is mutated in 3 cases, which the authors suggest is related to p53, however I could not find any analysis indicating that these were mutually exclusive of TP53 mutations (or if there is any relationship at all)

Overall, this is a large amount of work, but most of the conclusions are not novel and confirm prior studies. Unfortunately no functional data is provided related to the mutations observed. Finally, the

discussion is somewhat superficial, and does not even mention some other whole genome sequencing manuscripts (e.g. Rover-Bertrand et al.) Given the size of this cohort, it is suggested that greater insights may be derived by combining the data in this cohort with other datasets, or at least comparing the data.

We are grateful to the reviewers for their careful consideration of this manuscript and their helpful suggestions for improving the paper. We have addressed the reviewers' comments in full below. Reviewers' questions and comments are in bold black text and the authors' responses are in blue text.

Reviewer #1

Red in Figure 2 should be used for the more conventional copy number gain rather than loss to avoid reader confusion.

Colours in Figures 2 and 3 have been changed as suggested.

Reviewer #2

1) Title is not supported by evidence. If UVR signature is shown in iris UM, no causal link with the disease is provided. UVR signature also exists also in normal melanocytes or keratinocytes!

We accept that we do not show a causal link between UVR exposure and iris UM tumorigenesis, but we show for the first time molecular evidence this subset of UM have a UVR signature. We have modified the title accordingly.

2) L70 "One sample had noticeably more SVs, of which the majority (71%) were mid-sized (<100 kb) deletions, suggesting this was not due to chromothripsis" Limited information is provided for this interesting outlier case. Any correlation of driver mutations?

Analyses were done in relation to SVs and SNVs in the tumour and there were no notable features or driver mutations. We have now indicated this in the text.

3) L195 "In addition, two D3 tumours carried BAP1 mutations, indicating that although BAP1 inactivation typically occurs after M3, BAP1 aberration can also occur in D3 tumours, which may or may not later undergo loss of chromosome 3." These observations are not as surprising as they appear in the text. One is associated with a partial loss of chr3, which is relatively rare but previously reported,

The tumour in question does not have a partial loss of chr3, it is disomy 3.

and for the second case, BAP1 mutation seems to have emerged as a subclonal event (low VAF), in which the status of chr3 is yet to be determined.

It is likely correct that the nonsense mutation in this tumour is a subclonal event and we have thus added the following clarification in the main text:

"Of note, one of the mutations had a low variant allele frequency (VAF = 9/80) suggesting it was only present in a subclone and as copy number tools are not as sensitive as mutation callers, it is possible that the subclone had loss of heterozygosity (LOH) that was not detected by the algorithm."

4) Evidence for new drivers in UM are scarce:

- “three additional significantly mutated genes” but only one algorithm (Oncodrive-fm) shows this significance, and with a marginal p-value for CENPE.

It is not unexpected that SMGs are not always identified by all tools, since they focus on different criteria (which was our reason to run multiple tools), e.g. OncodriveCLUST is designed to find hotspot mutations, thus many tumour suppressors are missed; MutSigCV is more focused on the number of mutations, so genes with relatively few mutations are often missed; whereas Oncodrive-FM is more focused on the functional impact of mutations. These points are exemplified in our UM data by the observation that *BAP1*, *SF3B1*, and *GNA11* were each only called by only 2 of the 3 tools used.

- **CENPE: the second wild-type allele is retained with all variants**

We acknowledge that the *CENPE* disruption is heterozygous in these tumours. However there is evidence to support that *CENPE* disruption might function in a haploinsufficient manner. The discussion has been expanded to address and clarify this issue as follows: “Studies both in cell line and mouse models have demonstrated that CENP-E/Cenp-E functions in a haploinsufficient manner, with elevated levels of chromosome mis-segregation observed in heterozygous cells and animals (PMID: 17189716, PMID: 17974949). Further functional work on CENPE missense mutations is required to determine their impact in UM.”

L130 “It is possible that disruption of this pathway is responsible for creating genomic instability allowing for chromosome aberrations to occur.” The authors should have all data to answer this question. Any correlation with WGD ? with number of CNA ?

While UM with altered *CENPE* do have more CNA, this cannot be taken to indicate causation, as samples with many CNAs tend to have more LOH in virtually any gene – just like samples with high TMB tend to have more somatic mutations in any gene. We have added this correlation into the main text and also included the described caveat.

The authors should take into account that Oncodrive-fm does not correct for the size of the gene, and that CENPE is a very large gene (so it is highly likely that these variants are passenger).

Oncodrive-FM does correct for gene size. The algorithm computes the average FI score for all observed mutations in the gene, where the FI scores measure the functional impact of the mutations (e.g. from CADD). It next calculates the average score for the observed mutations and compares that with the distribution of averages when drawing mutations randomly. If the observed average score is more often higher than the average from randomisations, the gene is considered significantly mutated.

- **TP53: “Mutations in TP53 are common in many cancers but have not been described in UM to date.” False: see Hajkova et al Sci Rep 2018**

Hajkova et al. 2018 presented two cases with germline *TP53* variants (of unknown pathogenicity) proposed to be associated with Li-Fraumeni syndrome. What we report here is the common acquisition of somatic *TP53* mutations, the majority having confirmed pathogenicity.

and ref, Rodrigues et al CCR 2019.

We have referenced the one somatic *TP53* mutation described in Rodrigues et al. as well as a second case in the MSK-IMPACT data set; these observations were, however, single instances in cohorts of 42 and 43, respectively. Here we report for the first time that *TP53* is a statistically significantly mutated gene in UM. We have changed the text to: “Somatic mutations in *TP53* have been described in two UM. A hotspot mutation p.R175H ($n = 1216$ in IARC *TP53* database) was observed in a hypermutated metastatic UM with deficient MBD4 and another hotspot mutation p.M237I ($n = 196$ in IARC *TP53* database) was observed in a UM in a pan-cancer study of metastatic tumours. Here we identified *TP53* as an SMG and report six somatic *TP53* mutations across four tumours in addition to eight cases of LOH”

- RPL5 : None of these mutations are associated with LOH. One case is hypermutated.

We acknowledge that none of the UM has a double hit of *RPL5*, which is expected as *RPL5* has been described as a haploinsufficient tumour suppressor (Fancello *et al.* Oncotarget, 2017). We have extended the discussion on this point to make it clearer for the reader.

L151 “RPL5 along with RPL11 and ARF are responsible for inhibiting MDM2 and stabilising p53 in response to blocks in ribosome biogenesis and nucleolar stress, often triggered by oncogene-induced translational stress. Supporting the importance of this pathway in UM, TP53 and RPL5 were altered in 42% of cases in this cohort.” This is an overstatement supported by no evidence, but that the longest chromosome (chr1) is frequently altered in UM.

The discussion on *RPL5* has been rewritten and the text above has been deleted.

5) L167 “Interestingly, while iris melanomas are associated with earlier detection and favourable prognosis, 4 of 8 iris cases had M3, with two already having progressed to metastatic disease”. None of these metastatic patients received immunotherapy?

Unfortunately, neither of the two patients in the iris UM cohort who have developed metastases received immunotherapy – thus we cannot correlate response to treatment. We hope our report may lead to the analysis of previous and future metastatic iris UM patients who receive immunotherapy to determine if a correlation with improved response rates exists.

Minor comment

- The first publication of UM characterized by WGS (Furney Cancer Discov 2013) should be cited

We have added the Furney *et al.* reference.

- L97 “These mutations were generally mutually exclusive except for two PLCB4 p.D630 mutations. As seen in the UM TCGA cohort, overlap occurred with the minor GNAQ p.R183Q/ GNA11 p.R183C. hotspots rather than the stronger oncogenic p.Q209 hotspot mutations.” Confusing: something like “two PLCB4 p.D630 co-occurred with GNA p.R183” would be more straightforward.

Thanks for the suggestion. We have changed the main text to: “These mutations were generally mutually exclusive except for two PLCB4 p.D630 mutations that co-occurred with GNAQ/GNA11 p.R183H mutations. This co-occurrence between PLCB4 mutation and the minor hotspot p.R183,

rather than the stronger oncogenic p.Q209 hotspot mutations, has previously been described in the UM TCGA data.”

- L145: “These data suggest the two mutations are biallelic”. No, the authors suggest composite heterozygous mutations or mutations in trans.

We have changed the sentence to: “These data suggest the two mutations occurred on different alleles, with the majority of the transcripts from the p.R342* allele undergoing nonsense mediated decay.”

Reviewer #3

Multiple aberrations in TP53 pathway..... functional data for most of these mutations could be strengthened by evaluating recent, comprehensive analysis of p53 mutations (e.g. Giacornelli et al, Kotler et al, etc).

We thank the reviewer for this suggestion and have added the results from these characterisation studies into the discussion of our observed *TP53* mutations. For further clarity, we have also added a summary of these data together with data from the IARC TP53 database and the scores from the FATHMM prediction tool.

(4) RPL5 is mutated in 3 cases, which the authors suggest is related to p53, however I could not find any analysis indicating that these were mutually exclusive of TP53 mutations (or if there is any relationship at all).

We have added the following sentence about *RPL5* and *TP53* mutations: “Given the link between *RPL5* and *p53*, we tested for an association between aberration in *RPL5* and *TP53*. Indeed, mutations in these genes were mutually exclusive, but when also considering copy-number loss there was no association”.

The discussion is somewhat superficial, and does not even mention some other whole genome sequencing manuscripts (e.g. Rover-Bertrand et al.) Given the size of this cohort, it is suggested that greater insights may be derived by combining the data in this cohort with other datasets, or at least comparing the data.

We have extended the discussion substantially, particularly discussion the new SMGs. We have also extended the description of UM categories, how they correlate with the common UM mutations, and have compared our data with previous large cohorts such as TCGA and Royer-Bertrand et al.

REVIEWERS' COMMENTS:

Reviewer #2 (Remarks to the Author):

Johansson and Coll provide an analysis by WGS of the largest series of uveal melanoma (103 UM cases) described so far. In this revised manuscript, they propose some improvements and some modifications of their first manuscript, without providing any new key information. This analysis largely confirms previously published results, and further shows the presence of UVR mutational signature in iris UM. They demonstrate the rare but significant role of TP53 in this disease, which was previously reported. Evidence for RPL5 and CENPE as new drivers are minimal and unconvincing. It is difficult to understand the interest of using WGS rather than WES, (UVR role in iris melanoma is quite obvious using WES) when the authors are not exploiting their WGS data to extract additional information such as structural variants or alternations of non coding regions, such as promoter regions.

1) Title is not supported by evidence. If UVR signature is shown in iris UM, no causal link with the disease is provided. UVR signature also exists also in normal melanocytes or keratinocytes! We accept that we do not show a causal link between UVR exposure and iris UM tumorigenesis, but we show for the first time molecular evidence this subset of UM have a UVR signature. We have modified the title accordingly.

→ Thank you

2 L70 "One sample had noticeably more SVs, of which the majority (71%) were mid-sized (<100 kb) deletions, suggesting this was not due to chromothripsis" Limited information is provided for this interesting outlier case. Any correlation of driver mutations?

Analyses were done in relation to SVs and SNVs in the tumour and there were no notable features or driver mutations. We have now indicated this in the text.

→ Thank you

3) L95 "In addition, two D3 tumours carried BAP1 mutations, indicating that although BAP1 inactivation typically occurs after M3, BAP1 aberration can also occur in D3 tumours, which may or may not later undergo loss of chromosome 3." These observations are not as surprising as they appear in the text. One is associated with a partial loss of chr3, which is relatively rare but previously reported,

The tumour in question does not have a partial loss of chr3, it is disomy 3.

→ Indeed you classified MELA_0800 as disomic. However, this tumor has quite a low tumor content as estimated using GNAQ VAF (18%), so the chromosome status is questionable without further analyses. BAP1 VAF (11%) is thus compatible with a partial loss of chr3. Actually, genomic profiles of all tumors (such as FACETS) should be provided to clarify such outlier cases.

and for the second case, BAP1 mutation seems to have emerged as a subclonal event (low VAF), in which the status of chr3 is yet to be determined.

It is likely correct that the nonsense mutation in this tumour is a subclonal event and we have thus added the following clarification in the main text:

"Of note, one of the mutations had a low variant allele frequency (VAF = 9/80) suggesting it was only present in a subclone and as copy number tools are not as sensitive as mutation callers, it is possible that the subclone had loss of heterozygosity (LOH) that was not detected by the algorithm."

→ I do not share this analysis. The variant is moderate and no LOH is present in this tumor (BAP1 VAF 40% GNAQ 37%). So it is most likely a passenger mutation.

4) Evidence for new drivers in UM are scarce:

- "three additional significantly mutated genes" but only one algorithm (Oncodrive-fm) shows this significance, and with a marginal p-value for CENPE.

It is not unexpected that SMGs are not always identified by all tools, since they focus on different criteria (which was our reason to run multiple tools), e.g. OncodriveCLUST is designed to find hotspot mutations, thus many tumour suppressors are missed; MutSigCV is more focused on the number of mutations, so genes with relatively few mutations are often missed; whereas Oncodrive-FM is more focused on the functional impact of mutations. These points are exemplified in our UM data by the observation that BAP1, SF3B1, and GNA11 were each only called by only 2 of the 3 tools used.

- CENPE: the second wild-type allele is retained with all variants

We acknowledge that the CENPE disruption is heterozygous in these tumours. However there is evidence to support that CENPE disruption might function in a haploinsufficient manner. The discussion has been expanded to address and clarify this issue as follows: "Studies both in cell line and mouse models have demonstrated that CENP-E/Cenp-E functions in a haploinsufficient manner, with elevated levels of chromosome mis-segregation observed in heterozygous cells and animals (PMID: 17189716, PMID: 17974949). Further functional work on CENPE missense mutations is required to determine their impact in UM."

L130 "It is possible that disruption of this pathway is responsible for creating genomic instability allowing for chromosome aberrations to occur." The authors should have all data to answer this question. Any correlation with WGD ? with number of CNA ?

While UM with altered CENPE do have more CNA, this cannot be taken to indicate causation, as samples with many CNAs tend to have more LOH in virtually any gene – just like samples with high TMB tend to have more somatic mutations in any gene. We have added this correlation into the main text and also included the described caveat.

The authors should take into account that Oncodrive-fm does not correct for the size of the gene, and that CENPE is a very large gene (so it is highly likely that these variants are passenger). Oncodrive-FM does correct for gene size. The algorithm computes the average FI score for all observed mutations in the gene, where the FI scores measure the functional impact of the mutations (e.g. from CADD). It next calculates the average score for the observed mutations and compares that with the distribution of averages when drawing mutations randomly. If the observed average score is more often higher than the average from randomisations, the gene is considered significantly mutated.

→ Actually you are right. OncodriveFM does not correct for gene size but the use of FI score does the job. However, your p_value is very marginally significant (0.02) and for only one algorithm, and you are not taking into account that you tested all mutants (more than 3000 genes / more than 3000 tests) without any correction for multiple testing. Without any biological validation such as aneuploidy, your results are not convincing at all.

- TP53: "Mutations in TP53 are common in many cancers but have not been described in UM to date."
False: see Hajkova et al Sci Rep 2018

Hajkova et al. 2018 presented two cases with germline TP53 variants (of unknown pathogenicity) proposed to be associated with Li-Fraumeni syndrome. What we report here is the common acquisition of somatic TP53 mutations, the majority having confirmed pathogenicity.
and ref, Rodrigues et al CCR 2019.

We have referenced the one somatic TP53 mutation described in Rodrigues et al. as well as a second case in the MSK-IMPACT data set; these observations were, however, single instances in cohorts of 42 and 43, respectively. Here we report for the first time that TP53 is a statistically significantly mutated gene in UM. We have changed the text to: "Somatic mutations in TP53 have been described in two UM. A hotspot mutation p.R175H (n = 1216 in IARC TP53 database) was observed in a hypermutated

metastatic UM with deficient MBD4 and another hotspot mutation p.M2371 (n = 196 in IARC TP53 database) was observed in a UM in a pan-cancer study of metastatic tumours. Here we identified TP53 as an SMG and report six somatic TP53 mutations across four tumours in addition to eight cases of LOH"

→ OK

- RPL5 : None of these mutations are associated with LOH. One case is hypermutated.

We acknowledge that none of the UM has a double hit of RPL5, which is expected as RPL5 has been described as a haploinsufficient tumour suppressor (Fancello et al. Oncotarget, 2017). We have extended the discussion on this point to make it clearer for the reader.

→ OK

L151 "RPL5 along with RPL11 and ARF are responsible for inhibiting MDM2 and stabilising p53 in response to blocks in ribosome biogenesis and nucleolar stress, often triggered by oncogene-induced translational stress. Supporting the importance of this pathway in UM, TP53 and RPL5 were altered in 42% of cases in this cohort." This is an overstatement supported by no evidence, but that the longest chromosome (chr1) is frequently altered in UM.

The discussion on RPL5 has been rewritten and the text above has been deleted.

→ As for CENPE, your p_value for RPL5 is very marginally significant (0.02) and for only one algorithm, and you are not taking into account that you tested all mutants (more than 3000 genes / more than 3000 tests) without any correction for multiple testing. Without any biological validation such as aneuploidy, your results are not convincing at all.

5) L167 "Interestingly, while iris melanomas are associated with earlier detection and favourable prognosis, 4 of 8 iris cases had M3, with two already having progressed to metastatic disease". None of these metastatic patients received immunotherapy?

Unfortunately, neither of the two patients in the iris UM cohort who have developed metastases received immunotherapy – thus we cannot correlate response to treatment. We hope our report may lead to the analysis of previous and future metastatic iris UM patients who receive immunotherapy to determine if a correlation with improved response rates exists.

→ OK

Minor comment

- The first publication of UM characterized by WGS (Furney Cancer Discov 2013) should be cited. We have added the Furney et al. reference.

→ Thank you

- L97 "These mutations were generally mutually exclusive except for two PLCB4 p.D630 mutations. As seen in the UM TCGA cohort, overlap occurred with the minor GNAQ p.R183Q/ GNA11 p.R183C. hotspots rather than the stronger oncogenic p.Q209 hotspot mutations." Confusing: something like "two PLCB4 p.D630 co-occurred with GNA p.R183" would be more straightforward.

Thanks for the suggestion. We have changed the main text to: "These mutations were generally mutually exclusive except for two PLCB4 p.D630 mutations that co-occurred with GNAQ/GNA11 p.R183H mutations. This co-occurrence between PLCB4 mutation and the minor hotspot p.R183, rather than the stronger oncogenic p.Q209 hotspot mutations, has previously been described in the UM TCGA data."

→ Thank you

- L145: "These data suggest the two mutations are biallelic". No, the authors suggest composite heterozygous mutations or mutations in trans.

We have changed the sentence to: "These data suggest the two mutations occurred on different alleles, with the majority of the transcripts from the p.R342* allele undergoing nonsense mediated decay."

→ Thank you

Reviewer #3 (Remarks to the Author):

The authors have responded to most of the specific concerns raised in my prior review. However, along with Reviewer #1, I indicated that the results were not especially novel. This makes me less enthusiastic about the suitability of manuscript for Nature Communications.

We have addressed the reviewer's comments in full below. The Reviewer's original questions and comments are in black text, the Reviewer's new questions and comments are in bold black text, the authors' previous responses are in gray text, and the authors' current responses are in blue text.

3) L95 "In addition, two D3 tumours carried BAP1 mutations, indicating that although BAP1 inactivation typically occurs after M3, BAP1 aberration can also occur in D3 tumours, which may or may not later undergo loss of chromosome 3." These observations are not as surprising as they appear in the text. One is associated with a partial loss of chr3, which is relatively rare but previously reported,

The tumour in question does not have a partial loss of chr3, it is disomy 3.

→ **Indeed you classified MELA_0800 as disomic. However, this tumor has quite a low tumor content as estimated using GNAQ VAF (18%), so the chromosome status is questionable without further analyses. BAP1 VAF (11%) is thus compatible with a partial loss of chr3. Actually, genomic profiles of all tumors (such as FACETS) should be provided to clarify such outlier cases.**

and for the second case, BAP1 mutation seems to have emerged as a subclonal event (low VAF), in which the status of chr3 is yet to be determined.

It is likely correct that the nonsense mutation in this tumour is a subclonal event and we have thus added the following clarification in the main text:

"Of note, one of the mutations had a low variant allele frequency (VAF = 9/80) suggesting it was only present in a subclone and as copy number tools are not as sensitive as mutation callers, it is possible that the subclone had loss of heterozygosity (LOH) that was not detected by the algorithm."

→ **I do not share this analysis. The variant is moderate and no LOH is present in this tumor (BAP1 VAF 40% GNAQ 37%). So it is most likely a passenger mutation.**

It seems the reviewer has mixed up which of our responses belongs to which sample. Our first response was about MELA_0809, which was called disomy 3 – not partial loss of chr3. This sample has relatively high VAF in *BAP1* (40%) and *GNAQ* (37.5%). Our second response was about sample MELA_0800 for which the reviewer suggested the low *BAP1* VAF (9/80 = 11%) indicated a subclonal event. We agreed with this and added a note in the main text (see above). To make it clearer for the reader we have now added the sample name in that sentence: "Of note, one of these D3 tumours (MELA_0800) had a low *BAP1* variant allele frequency (VAF = 9/80) suggesting..."

The authors should take into account that Oncodrive-fm does not correct for the size of the gene, and that CENPE is a very large gene (so it is highly likely that these variants are passenger).

Oncodrive-FM does correct for gene size. The algorithm computes the average FI score for all observed mutations in the gene, where the FI scores measure the functional impact of the mutations (e.g. from CADD). It next calculates the average score for the observed mutations and

compares that with the distribution of averages when drawing mutations randomly. If the observed average score is more often higher than the average from randomisations, the gene is considered significantly mutated.

→ **Actually you are right. OncodriveFM does not correct for gene size but the use of FI score does the job. However, your p_value is very marginally significant (0.02) and for only one algorithm, and you are not taking into account that you tested all mutants (more than 3000 genes / more than 3000 tests) without any correction for multiple testing. Without any biological validation such as aneuploidy, your results are not convincing at all.**

We agree it is important to do a multiple test correction. We therefore did not use the p-values, but instead used the q-values to determine which genes were significantly mutated. The q-values are p-values adjusted for multiple testing following the Benjamini-Hochberg procedure. We have clarified that in the methods, explaining that q-values are Benjamini-Hochberg adjusted p-values and the sentence now reads: "A Benjamini-Hochberg adjusted p-value (q-value) below 0.05 was considered significant."

→ **As for CENPE, your p_value for RPL5 is very marginally significant (0.02) and for only one algorithm, and you are not taking into account that you tested all mutants (more than 3000 genes / more than 3000 tests) without any correction for multiple testing. Without any biological validation such as aneuploidy, your results are not convincing at all.**

Please see comment above.